# Spatial distribution of common childhood illnesses, healthcare utilisation and associated factors in Ethiopia: Evidence from 2016 Ethiopian Demographic and Health Survey

**Atkure Defar**[1,2]*, **Yemisrach B. Okwaraji**[1,3], **Zemene Tigabu**[4], **Lars Åke Persson**[1,3], **Kassahun Alemu**[2]

**1** Ethiopian Public Health Institute, Addis Ababa, Ethiopia, **2** Department of Epidemiology and Biostatistics, Institute of Public Health, College of Medicine and Health Sciences, University of Gondar, Gondar, Ethiopia, **3** London School of Hygiene and Tropical Medicine, London, United Kingdom, **4** Department of Paediatrics and Child Health, School of Medicine, College of Medicine and Health Sciences, University of Gondar, Gondar, Ethiopia

* atkuredefar@gmail.com

## Abstract

### Introduction

Childhood illnesses, such as acute respiratory illness, fever, and diarrhoea, continue to be public health problems in low-income countries. Detecting spatial variations of common childhood illnesses and service utilisation is essential for identifying inequities and call for targeted actions. This study aimed to assess the geographical distribution and associated factors for common childhood illnesses and service utilisation across Ethiopia based on the 2016 Demographic and Health Survey.

### Methods

The sample was selected using a two-stage stratified sampling process. A total of 10,417 children under five years were included in this analysis. We linked data on their common illnesses during the last two weeks and healthcare utilisation were linked to Global Positioning System (GPS) information of their local area. The spatial data were created in ArcGIS10.1 for each study cluster. We applied a spatial autocorrelation model with Moran's index to determine the spatial clustering of the prevalence of childhood illnesses and healthcare utilisation. Ordinary Least Square (OLS) analysis was done to assess the association between selected explanatory variables and sick child health services utilisation. Hot and cold spot clusters for high or low utilisation were identified using Getis-Ord Gi*. Kriging interpolation was done to predict sick child healthcare utilisation in areas where study samples were not drawn. All statistical analyses were performed using Excel, STATA, and ArcGIS.

### Results

Overall, 23% (95CI: 21, 25) of children under five years had some illness during the last two weeks before the survey. Of these, 38% (95%CI: 34, 41) sought care from an appropriate

**Data Availability Statement:** All the data related to this research article are available in this manuscript as text, tables, and figures. However, to replicate

our study findings, the EDHS datasets are in the public domain on the DHS measure survey website which is available at: https://dhsprogram.com/data/available-datasets.cfm (accessed 25 June 2019). The authors had no special access privileges to the DHS data, and other researchers will be able to access the data in the same manner as the authors using the provided URL link. Data access requests may also be sent to Bridgette Wellington (Data Archivist at The Demographic and Health Surveys (DHS) Program) at E-mail address: archive@dhsprogram.com.

**Funding:** The authors received no specific funding for this work.

**Competing interests:** The authors have declared that no competing interests exist.

provider. Illnesses and service utilisation were not randomly distributed across the country with a Moran's index 0.111, Z-score 6.22, P<0.001, and Moran's index = 0.0804, Z-score 4.498, P< 0.001, respectively. Wealth and reported distance to health facilities were associated with service utilisation. Prevalence of common childhood illnesses was higher in the North, while service utilisation was more likely to be on a low level in the Eastern, Southwestern, and the Northern parts of the country.

## Conclusion

Our study provided evidence of geographic clustering of common childhood illnesses and health service utilisation when the child was sick. Areas with low service utilisation for childhood illnesses need priority, including actions to counteract barriers such as poverty and long distances to services.

## Background

The Sustainable Development Goals (SDGs) emphasise the need for accelerated actions to reach the targets for improved maternal, newborn, and child health [1]. Every year, millions of children under five years of age die, mostly from preventable causes such as pneumonia, diarrhoea, and malaria. In almost half of the cases, malnutrition plays a role, while unsafe water, sanitation, and hygiene significantly contribute to these deaths. For these reasons, child mortality is not only an essential indicator of child health and well-being but also of overall progress towards the SDGs [2].

Sub-Saharan Africa continues to be the region with the highest under-five mortality rates and with high regional disparities in the occurrence of common childhood illnesses [3–5]. Approximately 1 in 11 children born in sub-Saharan Africa die before five years, i.e., nearly 15 times higher than the average of high-income countries [6].

Despite a high burden of common childhood illnesses in Ethiopia, the utilisation of health services is on a low level. According to the EDHS 2016, 7% of children under five years of age had symptoms of acute respiratory infection (ARI), i.e., cough and difficult breathing. Among these, only 31% sought treatment or advice. Fourteen percent had a fever, and 35% of those children sought care at a health facility. More than 1 in 10 children under the age of five had diarrhoea in the two weeks before the survey; 44% sought treatment or advice [7]. Without significant progress in this regard, the 2030 child health targets cannot be reached in time.

Many low- and middle-income countries are now implementing integrated community case management (iCCM) programmes, evaluating whether this strategy increase service coverage [8]. However, these efforts have challenges. Parents often describe difficulties when seeking care for their sick children. As shown in our previous paper, mothers of sick children experience cultural, social, and poverty-related barriers to care-seeking [9]. They may instead use traditional medicine or self-treatment before visiting a health facility [10]. Families' trust in their local health facility and service providers is reportedly often low [11].

There are several barriers to sick child care utilisation: inadequate referral practice, low quality of care, socioeconomic inequities, and challenging geography with hard-to-reach areas. These barriers may lead to geographic inequities or clustering of childhood illnesses and low care utilisation in specific areas [12–17]. Thus, based on the latest Ethiopian Demographic and Health Survey, we aimed to assess geographical differences in the occurrence of common childhood illnesses and the use of healthcare and analyse factors associated with service utilisation.

## Methods

### Study design, setting and period

This study used secondary data from the 2016 Ethiopian demographic and health survey (EDHS), which was a cross-sectional community-based study design.

Geographically, Ethiopia has nine regions (one more recently upgraded from zone–Sidama) and two administrative cities (Addis Ababa and Dire Dawa). Each region has urban and rural kebeles, which is the lowest administrative unit. More than 80% of the population live in three regions: Oromia, Amhara, and Southern Nations, Nationalities, and Peoples' Region (SNNPR). These regional states and two city administrations are sub-divided into 68 zones, 817 districts, and 16,253 kebeles. Ethiopia, 3˚-14˚N and 33˚ - 48˚E, is situated in the Eastern tip of Africa. The country covers an area of 1.1 million km$^2$ with geographical diversity, ranging from 4,550 meters above the sea level down to the Afar depression 110 metres below the sea level [18]. The survey was conducted from January 18 to June 27, 2016.

### Sampling

The 2016 Ethiopia Demographic and Health Survey (2016 EDHS) data analysed in this study, was the fourth Demographic Health Survey conducted in Ethiopia. The sample was designed to be nationally representative and provide estimates of key health and demographic indicators for the country as a whole and by urban-rural, region and administrative cities [7]. The sample was selected using a stratified two-stage cluster sampling design.

Each region was stratified into urban and rural areas, yielding 21 sampling strata. Samples of Enumeration Areas (EAs) were selected independently for each stratum. Implicit stratification and proportional allocation were achieved at each of the lower administrative levels (kebeles) by sorting the sampling frame within each sampling stratum before sample selection, according to administrative units in different levels, and by selecting with probability proportional to size at the first stage of sampling. This design allowed for specific indicators, such as childhood illness and care utilisation, to be calculated for each of Ethiopia's nine regional states and two city administrations. The 2007 Population and Housing Census, conducted by the central statistical agency, provided the sampling frame [18]. The census frame had a list of 84,915 EAs.

In the first stage, a total of 645 EAs (202 in urban areas and 443 in rural areas) were selected with probability proportional to EA size and with independent selection in each sampling stratum. A household listing operation was carried out in all of the selected EAs from September to December 2015. The resulting lists of households served as a sampling frame for the selection of households in the second stage. Some of the selected EAs were large, consisting of more than 300 households. To minimise the task of household listing, each large EA was segmented. Only one segment was selected for the survey with probability proportional to segment size created for the large EAs [7]. Household listing was conducted only in the selected segment; that implies that the 2016 EDHS cluster was either an EA or a segment of an EA. Sketch maps were drawn for each of the clusters, and all conventional households were listed.

In the second stage of selection, a fixed number of 28 households per cluster were selected with an equal probability systematic selection from the newly created household listing. The primary caregiver of children who were below five years of age at the time of the survey was interviewed about two-weeks occurrence of ARI (cough, and shortness or difficulty of breathing), fever and diarrhoea, and whether they had sought care from an appropriate facility or care provider. A total of 10,417 caregivers or mothers of under-five children were interviewed. The data were collected from January 18 to June 27, 2016.

## Survey instrument

The questionnaires were adapted from the model survey instruments developed for the MEASURE DHS project to reflect the population and health issues relevant to Ethiopia. This questionnaire is a standard tool used for collecting demographic and health related information in most countries. The English language version of the questionnaires were translated into three major local languages—Amharic, Oromiffa, and Tigrigna. The child module questionnaire was used to collect information about children under five years of age. The household questionnaire covered information on characteristics of the household, such as the source of water, type of toilet, floor material, and on the ownership of various durable goods. Mothers or other caretakers of under-five children were interviewed about child morbidity and healthcare utilisation during the two weeks before the survey [7].

## Variables

**Outcome measures.** The study outcomes were the two-week prevalence of common childhood illnesses, i.e., diarrhoea, fever, or cough plus shortness of breath or difficulty of breathing, and service utilisation for such illnesses. In the survey, the health status was assessed by answering the question to the mother: "Has your child had diarrhoea or cough or fever in the last two weeks?", followed by probing questions in the case of respiratory problems. The mothers or caregivers were asked if they sought care or advice for their sick children from defined governmental or non-governmental health facilities or providers (**Table 1**).

## Exposure measures and co-variates

Table 1 shows the outcomes and potential predictors for health service utilisation for common childhood illness used in the analysis; age and sex of the child, woman-headed household, age of the mother or caregiver, mothers' or caregivers' education level, household size, wealth index, and distance to health facility reported as a problem.

## Data management and analysis

The dataset was downloaded from the Measure DHS website (www.measuredhs.com) after becoming an authorized user (AuthLetter 130884). Data included the GPS location (latitude

**Table 1. Description of variables used in this study.**

| Variables | Cluster level measurements | Variable Type |
|---|---|---|
| Common childhood illness | The proportion of children aged 2–59 months with fever, diarrhoea, or suspected pneumonia in the last two weeks prior to the survey | Outcome |
| Sick childcare utilisation | The proportion of children with common childhood illnesses, for whom care was sought at health posts, health centres, hospitals, clinics or providers | Outcome |
| Child age | Average age of the children (in months) | Explanatory |
| Sex of the child | Proportion of male children | Explanatory |
| Wealth | Proportion of households in the highest (richest) quintile | Explanatory |
| Head of the household | Proportion of female headed households | Explanatory |
| Literacy | Proportion of literate mothers | Explanatory |
| Age of the mother | Average mothers' age (in years) | Explanatory |
| Household size | Average number of household members | Explanatory |
| Distance | Proportion of women who said distance to a health facility is not a big problem | Explanatory |

and longitude) of the enumeration areas. We did a descriptive analysis of socio-demographic characteristics, household wealth and sick child service utilisation at cluster level. Data were analysed using STATA v14 statistical software.

A spatial analysis was applied to detect geographic variation in the two-week prevalence of common childhood illnesses, health service utilisation and association between selected variables and heath service utilisation, the shapefiles are freely accessed using the fowling link: https://africaopendata.org/dataset/ethiopia-shapefiles and maps are produced using ArcGIS version 10.5 (ESRI, Redlands, CA, USA). Spatial dependency of the outcomes were assessed at cluster level using Moran's index (Global) value. The spatial autocorrelation analysis estimated Moran's index and z-score, and its associated p-values for observed and expected index values, given the number of features and variance. A positive Moran's index indicates a tendency towards clustering, while a negative Moran's index indicates a tendency towards dispersion.

Spatial heterogeneity of significantly high or low of common childhood illnesses and healthcare utilisation was computed for each cluster using the Getis-Ord Gi*-statistic tool in ArcGIS, which estimates a z-score of the observed and expected spatial clustering with high or low values and associated p-values. The result of Getis-Ord Gi* tells us how strong the linear model is within and between geographic areas and identifies which areas that contribute to a high or low degree in the spatial regression modelling. In this process, a value of 1 in the inverse distance conceptualisations of spatial relationships was used to avoid computation of the default threshold and row standardization was applied. A positive z-score with P-value <0.05 indicates clustering of hotspots of the outcomes, whereas a negative z-score with p-value of <0.05 indicates clustering of low spots.

We performed Ordinary Least Square (OLS) analysis to assess the relationship between different explanatory variables and health service utilisation. Multicollinearity was checked in the final set of variables using variance inflation factors (VIF), where VIFs greater than 10 indicated multicollinearity [19].

We spatially interpolated values at points without measurements based on the values at known points. The Kriging spatial interpolation method was used for predicting sick childcare utilisation in areas without measurements since it had a small mean square error and residual. The interpolation produced smoothed maps of sick child health service utilisation by predicting the proportion of sick child health service utilisation in the un-sampled locations (enumeration areas). The general formula for interpolation is formed as a weighted sum of the data:

$$\hat{Z}(s_0) = \sum_{i=1}^{N} \lambda_i Z(s_i)$$

Where:
$Z(s_i)$ = the measured value at the i$^{th}$ location
$\lambda_i$ = an unknown weight for the measured value at the i$^{th}$ location
$s_0$ = the prediction location
N = the number of measured values

## Results

### Characteristics of the study participants

The mothers or caregivers of 10,417 children under the age of five years were included in this analysis (Table 2). Two fifth, (40%) of the children were less than 24 months of age. The mean age of the caregivers was 29.6 years. Two-thirds (65%) of the mothers had no education, more than half were Muslims (**Table 2**).

**Table 2. Sociodemographic characteristics of respondents of children under the age of five in Ethiopia, 2016.**

| Characteristics | Frequency | Percentage |
|---|---|---|
| Sex of child | | |
| Male | 5483 | 51.5 |
| Female | 5158 | 48.5 |
| Current age of child (months) | | |
| <6 months | 1137 | 10.7 |
| 6–11 months | 1069 | 10 |
| 12–23 months | 2036 | 19.1 |
| 24–35 months | 2047 | 19.2 |
| 36–47 months | 2087 | 19.6 |
| 48–49 months | 2265 | 21.3 |
| Region | | |
| Tigray | 1033 | 9.7 |
| Afar | 1062 | 10 |
| Amhara | 977 | 9.2 |
| Oromia | 1581 | 14.9 |
| Somali | 1505 | 14.1 |
| Benishangul-Gumz | 879 | 8.3 |
| SNNP | 1277 | 12 |
| Gambela | 714 | 6.7 |
| Harari | 605 | 5.7 |
| Addis Adaba | 461 | 4.3 |
| Dire Dawa | 547 | 5.1 |
| Mother/caregivers' age (years) | | |
| Mean ± SD | 29.6 ± 6.6 | |
| Mothers'/caregivers' educational level | | |
| No education | 6838 | 64.3 |
| Primary | 2678 | 25.2 |
| Secondary | 734 | 6.9 |
| Higher | 391 | 3.7 |
| Religion | | |
| Orthodox | 3082 | 29 |
| Catholic | 72 | 0.7 |
| Protestant | 1862 | 17.5 |
| Muslin | 5442 | 51.1 |
| Other | 183 | 1.7 |
| Current marital status | | |
| Never in union | 61 | 0.6 |
| Married | 9903 | 93.1 |
| Living with partner | 105 | 1 |
| Widowed | 135 | 1.3 |
| Divorced | 328 | 3.1 |
| No longer living together/separated | 109 | 1 |
| Wealth index quintiles | | |
| 1st (lowest) | 3993 | 37.5 |
| 2nd | 1782 | 16.7 |
| 3rd | 1466 | 13.8 |
| 4th | 1308 | 12.3 |

(*Continued*)

**Table 2.** (Continued)

| Characteristics | Frequency | Percentage |
|---|---|---|
| 5[th] (highest) | 2092 | 19.7 |
| Household has electricity | | |
| No | 8141 | 77.5 |
| Yes | 2367 | 22.5 |
| Type of toilet use | | |
| Unimproved | 8718 | 83 |
| Improved | 1790 | 17 |
| Source of drinking water | | |
| Unimproved | 4005 | 38.1 |
| Improved | 6503 | 61.9 |
| Perceived distance to a health facility | | |
| Big problem | 5834 | 54.8 |
| Not a big problem | 4807 | 45.2 |
| Mother/caregiver is head of household | | |
| No | 8836 | 83 |
| Yes | 1805 | 17 |

## Common childhood illness and health service utilisation

Seven percent of children under the age of five had symptoms of ARI in the two weeks before the survey (**Table 3**). Of these, three out of ten sought care from an appropriate facility or provider. Fourteen percent of children under the age of five were reported to have fever in the two weeks before the survey. Care from a health facility or provider was sought only for 36% (95% CI: 32, 40). Twelve percent of children under the age of five had diarrhoea in two weeks before the survey. More than four out of ten children under the age of five who had diarrhoea had sought treatment. Fig 1 shows the distribution of common childhood illnesses and sick child health service utilisation. All except the green dots in the map indicate study clusters with at least one sick child with fever, diarrhoea or suspected pneumonia. The 94 clusters without sick children and 18 clusters from Somali region without shape files were excluded in the spatial analysis. Therefore, the results of the spatial analysis are based on 531 clusters.

In the spatial autocorrelation analysis, we estimated the distribution of the prevalence of children with common childhood illnesses and associated health service utilisation. Fig 2 depicts that the prevalence of sick children at cluster level were close to each other—there was less than 1% (99% degree of confidence) likelihood that this clustered pattern could be random

**Table 3. Proportion of children who had symptoms of ARI, fever, and diarrhoea in the two weeks prior to the survey in Ethiopia, 2016.**

| Reported illness in the last two weeks prior to the survey | Number | Prevalence of illness (95% CI) | % of children sought care at the facility (95% CI) |
|---|---|---|---|
| Illness in the last 2 weeks | | | |
| Acute respiratory infection | 10,417 | 6.63 (5.63, 7.79) | 29.81* |
| Fever | 10,417 | 14.35 (13.09, 15.71) | 35.74 (32.16, 39.49) |
| Diarrhoea | 10,417 | 11.78 (10.6, 13.08) | 42.56* |
| Common childhood illness (diarrhoea, fever or acute respiratory illness) | 10,417 | 23.08 (21.36, 24.88) | 37.38 (34.03, 40.85) |

*lack adequate number of primary sampling units or observation per stratum to estimate the standard error to get the 95% CI for acute respiratory infection and diarrhoea.

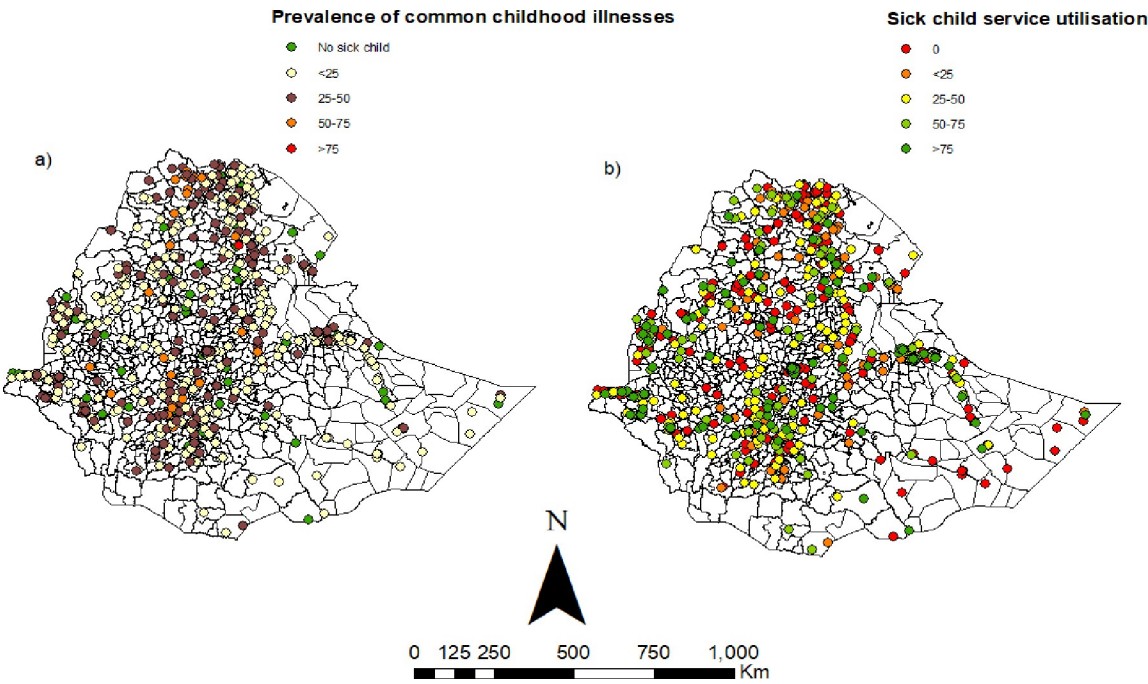

**Fig 1. The geographic distribution of a) the prevalence of common childhood illnesses and b) sick child health service utilisation in Ethiopia 2016.** *The shapefiles are freely accessed using the fowling link*: *https://africaopendata.org/dataset/ethiopia-shapefiles*.

(*Moran's index 0.111, Z-score 6.22, P<0.001*). The corresponding pattern for sick child service utilisation was also not randomly distributed (*Moran's index = 0.0804, Z-score 4.498, P< 0.001*) (**Fig 2**).

## Hotspot analysis of the prevalence of common childhood illnesses and service utilisation

As shown in Fig 3, the hot spot areas for common childhood illnesses were found in the Northern and central parts of the country, while the cold spots were in the Eastern and Western

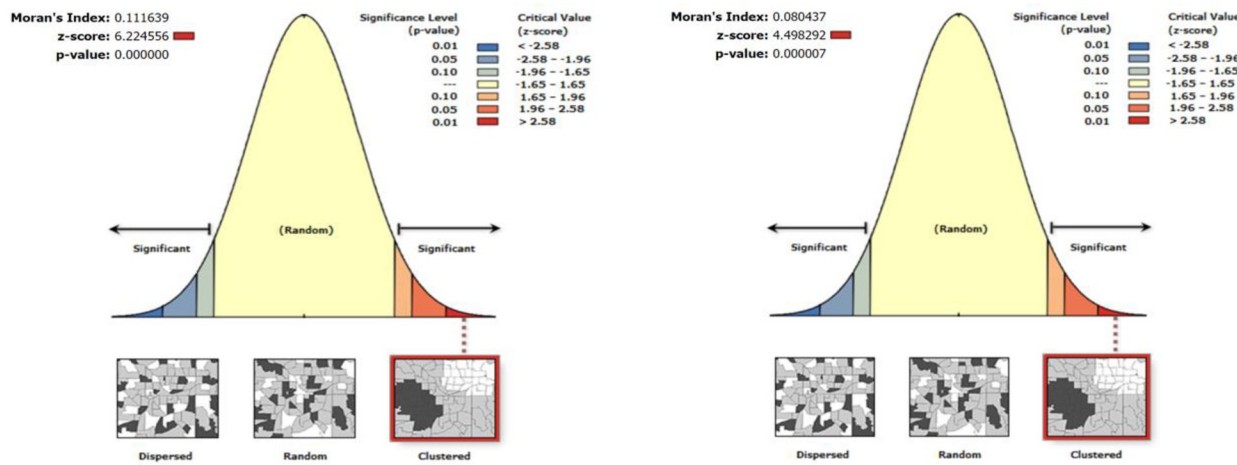

**Fig 2. Spatial patterns of sick child and health service utilisation for common childhood illnesses in Ethiopia 2016.** Left to right, the graphs show 1) distribution for common childhood illness 2) distribution of sick child health service utilisation.

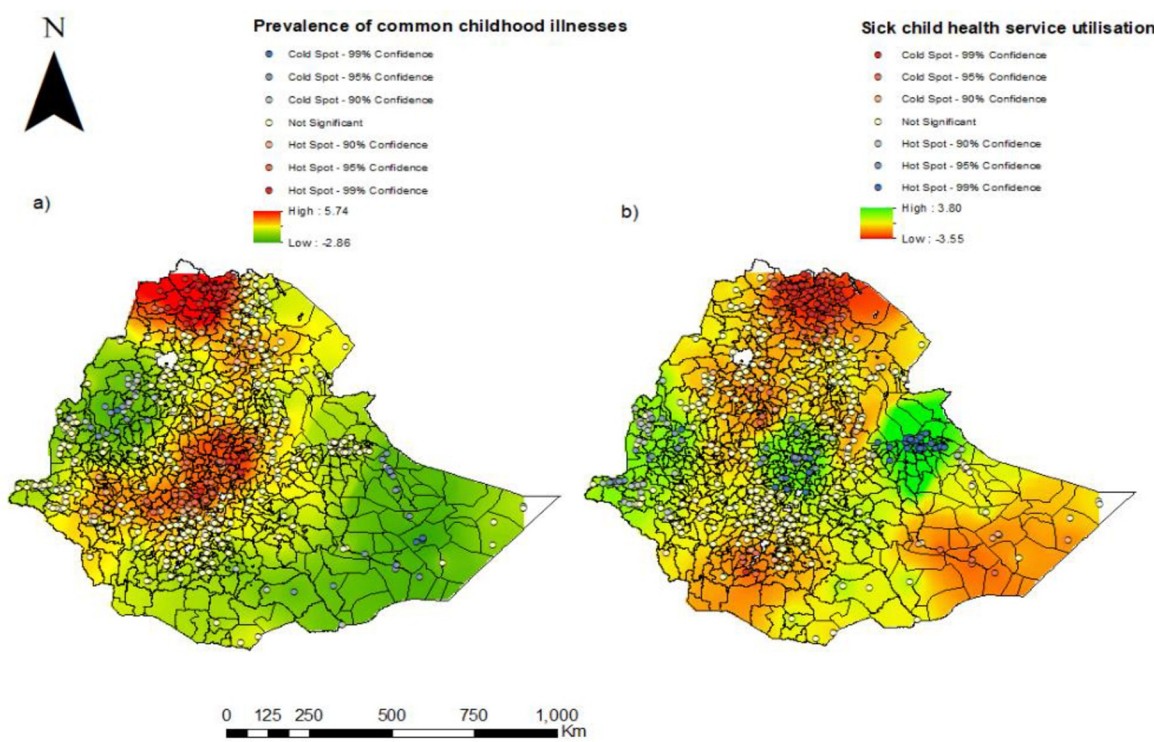

**Fig 3. Hotspot analysis for the prevalence of common childhood illnesses and child health service utilisation in Ethiopia 2016.** The shapefiles are freely accessed using the fowling link: https://africaopendata.org/dataset/ethiopia-shapefiles.

parts. The hotspot clusters of care utilisation were in the central part; Addis Ababa, Dire Dawa, Harari and the Western parts, whereas the cold spots were found in the North. In Fig 3, the yellow circles indicate non-significant clusters of care utilisation.

Fig 4 shows the cluster and outlier analysis using Anselin Local Morans I. The highest coverage of services was found in Harari region and the two city administrations, i.e., Dire Dawa, Addis Ababa and a few parts of Benishangul-Gumz. In those areas, we found 53 clusters with high values surrounded by other high values. Twenty-eight clusters in Tigray and Somali regions had low care utilisation, surrounded by low values. The number of non-significant and mixed clusters are shown in Table 4.

## Interpolation of prevalence of common childhood illnesses and care utilisation

Kriging interpolation showed high illness prevalence in the North and a few areas in central and Western parts of the country. Service utilisation was low in Eastern, South-Western and the North-Eastern parts of the county. Service utilisation was predicted to be high in Central, Western, and North-Eastern parts (**Fig 5**).

## Factors associated with health service utilisation for common childhood illnesses

We explored possible determinants for clustering of the coverage of sick childcare utilisation using a global spatial regression model and it has shown in Table 5. Household wealth and mothers' reported problem with distance to health facilities were associated with care utilisation. Other variables, i.e., age of the mother or the child, female head of the households,

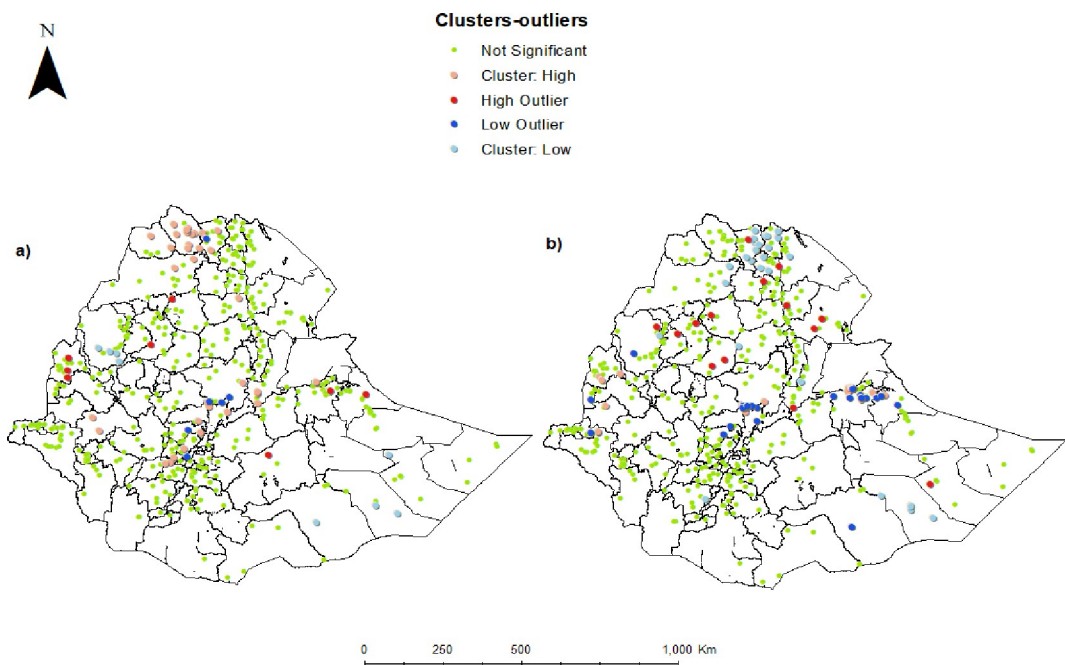

**Fig 4. Spatial cluster and outlier for a) common childhood illness and b) sick childcare utilisation for common childhood illnesses in Ethiopia 2016.** *The shapefiles are freely accessed using the fowling link*: *https://africaopendata.org/ dataset/ethiopia-shapefiles*.

number of household members, and education level were not statistically significantly associated with service utilisation. Spatial autocorrelation analysis of the residual was done to check the model stability. The pattern of the residual was not significantly different from a random distribution (Moran's index -0.075, p-value = 0.345) (**Table 5**).

## Discussion

Almost a quarter of the Ethiopian children in the 2016 Demographic and Health Survey had suffered from common illnesses during two weeks before the survey. The illnesses included

**Table 4. Spatial clustering (high, low, or non-significant levels) of common childhood illnesses and care utilisation.** Geographic areas and number of clusters included. Ethiopian 2016 Demographic and Health Survey.

| Indicator | Clustering | | | |
|---|---|---|---|---|
| | **High-high** | **Low-high or high-low** | **Low-low** | **Non-significant** |
| Prevalence of common childhood illnesses | Tigray, Central Ethiopia 41 | Amhara, Addis Ababa, Diredawa 14 | Somali, Benishangul-Gumz 8 | 468 |
| Sick child care utilisation | Harari, Dire Dawa, Addis Ababa, Benishangul-Gumz 53 | Dire Dawa, Addis Ababa, Central Ethiopia 42 | Tigray, Somali 28 | 408 |

High-high = clusters with high levels surrounded by high-level clusters.

Low-high = clusters with low levels surrounded by high-level clusters.

High-low = clusters with high levels surrounded by low-level clusters.

Low-low = clusters with low levels surrounded by low-level clusters.

Not significant = Areas with no statistically significant spatial autocorrelation (p<0.05).

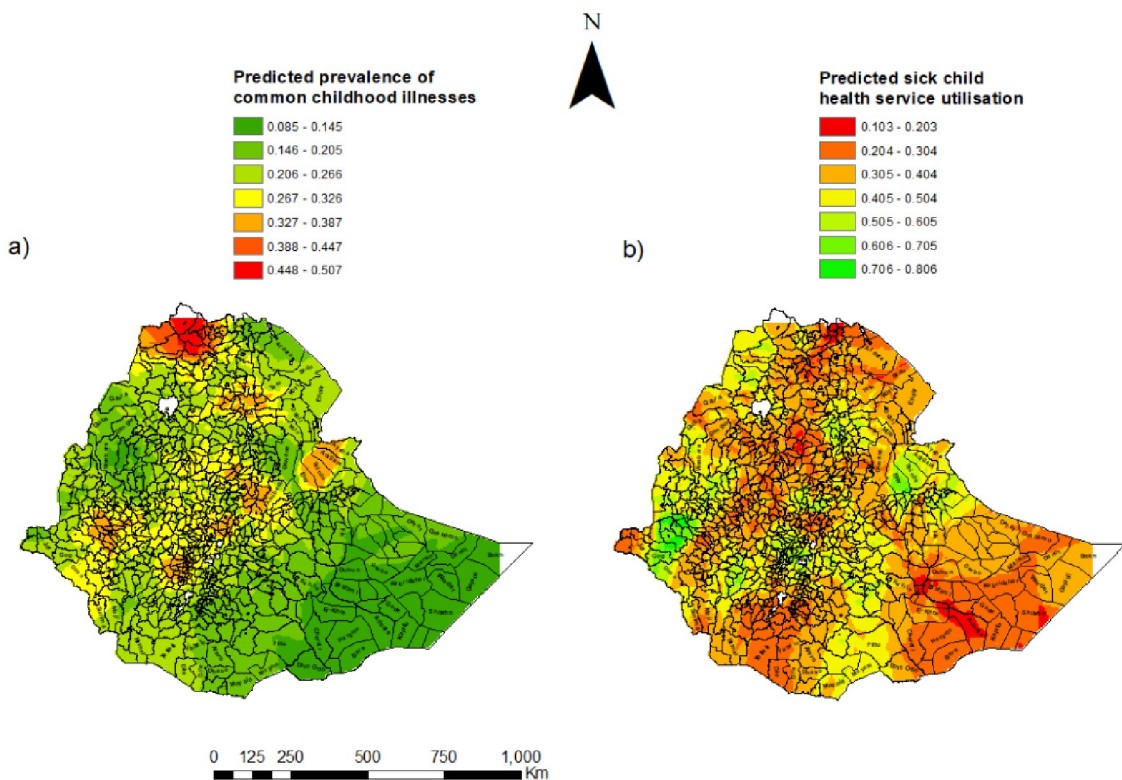

**Fig 5. Kriging interpolation of the spatial clustering of common childhood illness and sick child health service utilisation in Ethiopia 2016.** *The shapefiles are freely accessed using the fowling link*: *https://africaopendata.org/dataset/ethiopia-shapefiles*.

fever, diarrhoea and symptoms of acute respiratory infection. Of these sick children, 37% had sought care from an appropriate healthcare provider at health posts, health centres, hospitals, clinics or private clinics. The spatial analyses indicated geographic inequities in the occurrence of these common childhood diseases. The care utilisation for common childhood illness also showed such patterns. The highest coverage of health services for sick children was in Harari region and the two city administrations Dire Dawa and Addis Ababa. In contrast, the lowest level of care utilisation was found in Tigray and Somali regions. Children in the wealthiest households were more likely to receive care from an appropriate provider. Also, women who

**Table 5. Factors associated with sick child health service utilisation.** Adjusted ordinary least square analysis.

| Predictors at cluster level | Estimate | Std.error | t-value | P value | vif* |
|---|---|---|---|---|---|
| Intercept | 0.676614 | | | | |
| Average age of the children (months) | -0.002 | 0.003 | -0.774 | 0.439 | 1.079 |
| Proportion of household in the highest/richest quintile in the cluster | 0.165 | 0.058 | 2.807 | 0.005 | 2.92 |
| Proportion of female-headed households in the cluster | 0.032 | 0.074 | 0.430 | 0.667 | 1.080 |
| Proportion of literate mothers in the cluster | 0.026 | 0.076 | 0.339 | 0.734 | 2.861 |
| Mothers' age in years (average) | -0.005 | 0.006 | -0.846 | 0.3975 | 1.274 |
| Average number of household members | -0.025 | 0.015 | -1.598 | 0.110 | 1.493 |
| Proportion of women who said distance to health facility is not a big problem | 0.196 | 0.054 | 3.589 | 0.000 | 1.648 |

*indicates the amount of multicollinearity in the regression variables.

reported that distance was not a problem to access health facilities were more likely to have taken their children for treatment or advice.

The two-week prevalence of common illnesses of 23% was higher than reported in the studies conducted in 46 districts of the four regions of Ethiopia in 2018 and 2020 [14,20]. Child illness recall interviews are prone to interview biases, and more or less probing can influence these levels. Moreover, the DHS methodology is standardised and results are judged to be relatively reliable.

Our study showed geographical clustering in the prevalence of common childhood illnesses. Similarly, a study conducted in North-West Ethiopia showed spatial clustering of childhood diarrhoea [15]. Contrary to our study, a household-based study conducted in Amhara region, North-Western Ethiopia did not show any geographic variation in the prevalence of childhood diarrhoea [21]. The reason could be seasonal variation and other study area characteristics. Our study focused on all common childhood illnesses, while the mentioned study was limited to diarrhoea with its strong seasonality.

We found that 37% of the sick children had sought care from an appropriate provider. This level of care-seeking is close to the study conducted 2016 in 46 districts of four Ethiopian regions, which showed 35% [14]. Care utilisation was found to be low, 27%, in study conducted in Addis Ababa [22]. Such differences could be related to design and sample; our study was a community-based survey, while the latter was facility-based, which is prone to selection bias.

Our study showed clustering of child healthcare utilisation for common childhood illnesses. There were similar geographic patterns of care-seeking across countries in a sub-Saharan African study [23]. Our current results differ from the findings reported from our previous study in forty-six districts of four Ethiopian regions, which showed no evidence for geographic clustering in the utilisation of health services for common childhood illnesses [20]. Several interventional activities have been undergoing at primary health care level that could reduce morbidity and improve the child health service utilization since 2016 onwards could explain the difference partly.

Spatial clustering has been observed in other child health services, such as immunisation, in a review based on 183 surveys from 2000 to 2016, including data from 881,268 children in 49 African countries. It showed substantial geographical inequalities in DPT immunisation coverage across and within African countries [24]. The way immunization is organised–only at fixed sites or also with outreach services–may be one reason for this variation [25].

The coverage of care-seeking may increase as the integrated community case management programme is enhanced. In this programme, community health workers diagnose and treat pneumonia, diarrhoea, malaria and other common illnesses. These efforts could reduce geographic barriers to care seeking. However, problems with low quality of care remain a challenge [26].

Care seeking for sick children may be associated with the child's age, mother's education, family size, previous experiences of illnesses, and family history of under-five child death [22]. In our study, we found that household wealth and experience of distance from home to health facility were associated with care seeking. An analysis of data from the EDHS 2011 showed household wealth and maternal education to be associated with care seeking [27]. These associations did not appear in the study conducted in four regions of the country where we investigated the association between distance from the household to the nearest health facility, household wealth and care seeking [14,17].

Ordinary Kriging and empirical Bayesian Kriging are powerful interpolation methods that optimize the weight [28]. The Prediction of the proportion of sick child health service utilisation in the un-sampled locations (enumeration areas) showed that, the estimated occurrence

of common childhood illness was vary by areas (high in the North, central, and Western). It is predicated that low for sick child health service utilisation in Eastern, South-western and the North-eastern parts of the county. In contrast, care utilisation was more likely high in central, Western, North-eastern parts.

## Strengths and limitations

The study was based on nationally representative data and sampled to provide estimates for country and regions. Although the number of clusters were reduced by 94 from the original plan, as there was no shape files for them and data on illnesses were based on mothers' recall, the methodology used is established and considered to produce reliable data.

We used spatial statistics method- Spatial Autocorrelation and hotspot analysis that are highly efficient in detecting local clusters with good accuracy to provide valuable information about spatial disparity of childhood illnesses and childcare utilisation that may be relevant to provoke further investigation in the study area [29–31].

We also used a robust kriging interpolation technique that yields best extrapolation using the values (proportions of sick children and service coverage) of each clusters, not only the location.

## Conclusion

The geographic distribution of common childhood illness and health service utilization for common childhood illness in Ethiopia showed geographic inequities. Wealth and distance were associated with the health care utilisation for common childhood illnesses. Some areas of the country had very low service utilisation. As such, the geographical analysis could pinpoint areas of high need to inform decisions about resource allocation. There is a need to focus on community-level interventions aimed at enhancing equity in child health. Identifying hot and cold spots can inform targeted interventions and strategies to promote universal health coverage.

## Acknowledgments

We would like to acknowledge the Measure DHS for allowing free access to the data. The authors would like to thank Mr Alemayehu Hussen for his consultation and support in the data management and analysis.

### Ethics approval and consent to participate

Although the study used a secondary data, a formal procedure was followed to access the data, we obtained approval from the Demographic and Health Surveys program to access the data (Ref number: AuthLetter_130884). In addition, the University of Gondar has provided ethical approval (O/V/P/RCS/05/214/2018) to conduct this analysis. We used de-identified data (summary data without individuals identity) to ensure confidentiality. We followed the international standard of strengthening the reporting of observational studies in epidemiology (STROBE) [32]. Maps were created by study investigators using open access data sources.

## Author Contributions

**Conceptualization:** Atkure Defar, Yemisrach B. Okwaraji, Zemene Tigabu, Lars Åke Persson, Kassahun Alemu.

**Formal analysis:** Atkure Defar, Lars Åke Persson, Kassahun Alemu.

**Investigation:** Atkure Defar.

**Methodology:** Atkure Defar, Yemisrach B. Okwaraji, Zemene Tigabu, Lars Åke Persson, Kassahun Alemu.

**Resources:** Kassahun Alemu.

**Supervision:** Yemisrach B. Okwaraji, Zemene Tigabu, Kassahun Alemu.

**Writing – original draft:** Atkure Defar, Lars Åke Persson, Kassahun Alemu.

**Writing – review & editing:** Atkure Defar, Yemisrach B. Okwaraji, Zemene Tigabu, Lars Åke Persson, Kassahun Alemu.

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
