## [Decision Letter · Decision Letter 0]

2 Jun 2022

PONE-D-22-12369Spatial distributions of common childhood illnesses, healthcare utilisation and associated factors in Ethiopia: Evidence from 2016 Ethiopian Demographic and Health SurveyPLOS ONE

Dear Dr. Defar,

Thank you for submitting your manuscript to PLOS ONE. After careful consideration, we feel that it has merit but does not fully meet PLOS ONE’s publication criteria as it currently stands. Therefore, we invite you to submit a revised version of the manuscript that addresses the points raised during the review process. Please submit your revised manuscript by Jul 17 2022 11:59PM. If you will need more time than this to complete your revisions, please reply to this message or contact the journal office at plosone@plos.org. Please include the following items when submitting your revised manuscript:A rebuttal letter that responds to each point raised by the academic editor and reviewer(s). You should upload this letter as a separate file labeled 'Response to Reviewers'.A marked-up copy of your manuscript that highlights changes made to the original version. You should upload this as a separate file labeled 'Revised Manuscript with Track Changes'.An unmarked version of your revised paper without tracked changes. You should upload this as a separate file labeled 'Manuscript'.

We look forward to receiving your revised manuscript.

Kind regards,

Atalay Goshu Muluneh, MPH

Academic Editor

PLOS ONE

Journal Requirements:

 [This study used secondary data from the Ethiopian Demographic Health survey. No specific funding was received for the study.] 

5 .We note that Figures 1-5 in your submission contain map/satellite images which may be copyrighted. All PLOS content is published under the Creative Commons Attribution License (CC BY 4.0), which means that the manuscript, images, and Supporting Information files will be freely available online, and any third party is permitted to access, download, copy, distribute, and use these materials in any way, even commercially, with proper attribution. For these reasons, we cannot publish previously copyrighted maps or satellite images created using proprietary data, such as Google software (Google Maps, Street View, and Earth). For more information, see our copyright guidelines: http://journals.plos.org/plosone/s/licenses-and-copyright.

a) You may seek permission from the original copyright holder of Figures 1-5 to publish the content specifically under the CC BY 4.0 license.  

6. We note you have included a table to which you do not refer in the text of your manuscript. Please ensure that you refer to Table 5 in your text; if accepted, production will need this reference to link the reader to the Table.

Additional Editor Comments:

Dear Authors of Spatial distributions of common childhood illnesses, healthcare utilisation and associated factors in Ethiopia: Evidence from 2016 Ethiopian Demographic and Health Survey

You welcome to consider PLOS ONE for your scientific work. Rigorous scientific review is very important step in science. Your manuscript requires major revision.

Academic editor’s comment

1. PLOS ONE requires data used for this scientific work. Even though you used freely available national data, you are requested to attach the specific data set you used for this analysis.

2. Add key-words from the abstract

3. We saw flaws, and different grammatical and typological errors, hence language edition with experts is required.

4. Please strictly follow the PLOS ONE journal guideline (insert line number for the manuscript, add key words…)

Reviewer 1 comment

- There is a 2020 EDHS report, authors should use more recent data for analysis as changes might of occurred in the country that might have improved or increased prevalence of childhood diseases and utilization of services

- The section on study design was used to describe Ethiopia instead of telling us the type of study, the target population. The study would benefit from a revision of this section.

- What do the authors mean by “probability proportional to segment size”.

- “A total of 10,417 caregivers or mothers of under-five children were interviewed” contradicts the sample size information in the abstract and the result section “10,641”

- “Sick” or “Seek”. I observed the use of the word sick a couple of times. Please correct the word as appropriate

- The term “the two-week prevalence of common childhood illnesses “is not clear. Am not sure this is the term used in the DHS report. Authors need to ensure consistency of terms compared to the EDHS report.

- “Table 1 shows the outcomes and potential predictors for health service utilization”. Table 1 didn’t differentiate between outcomes and predictors. Update table 1 to contain all these variables “age and sex of the child, woman-headed household, age of the mother or caregiver, mothers’ or caregivers’ education level, household size, wealth index, and distance to health facility”.

- This statement “The 94 clusters without sick children and 18 clusters from Somali region without shape files were excluded in the spatial analysis. Therefore, the results of the spatial analysis are based on 531 clusters” however it should be a part of methodology as a limitation of the study

- In the result section, I had expected to see the prevalence of each of the childhood disease which were mentioned in the methodology section however am seeing a description of the prevalence. Authors should endeavor to present the prevalence of the childhood disease

- “*lack adequate number of primary sampling units or observation per stratum to estimate the standard error to get the 95% CI for acute respiratory infection and diarrhoea”. Are you saying only one primary sampling unit had people seeking health care? I think authors should endeavor to check this analysis to ensure accuracy.

- Table 4 seems to be a description of measures. I believe this should be a part of the methodology section

- The figure 1 maps aren’t correct. Prevalence is percentage vales and should represent an area. Dot distribution analysis can’t be used to represent a prevalence analysis. Authors should replot the maps

- Figure 2 don’t represent spatial patterns. Please change the distribution curves to maps. This analysis is not properly carried out

Reviewer 3:

From the title remove S from the word Distributions, it should be distribution

Title: Spatial distributions of common childhood illnesses, healthcare utilisation and associated

factors in Ethiopia: Evidence from 2016 Ethiopian Demographic and Health Survey

Abstract:

Methods: From the second line “A total of 10,641 children under the age of five years were included in this analysis” so remove “the age of”

Change the phrase interpolation kriging by Kriging interpolation

Background

At the fourth sentence: As we previously have shown, mothers of sick children experience cultural, social, and poverty-related barriers to care-seeking. Replace this sentence by “As shown in our previous paper, mothers of sick children experience cultural, social, and poverty-related barriers to care-seeking”

Methods

Sampling: From the 1st sentence please remove the comma and phrase “which we”

Outcome measurement section: replace the phrase “In the survey, the health status was assessed by the answering to the question to the mother:” by In the survey, the health status was assessed by answering the question to the mother:

Data management part third paragraph 1st sentence replace the phrase “Spatial heterogeneity of significant high or low of common” by Spatial heterogeneity of significantly high or low common..)

Result

The 1st sub heading please replace the word characteristic by characteristics

As you know the result section should focus on what researchers found while the methods elaborate on what we did or what procedures we used. The following sentences should be part of methods not results section “Using hot spot analysis (Getis-Ord Gi*), we classified each cluster as being part of a spatial clustering with hotspot, cold spot at 90%, 95%, or 99% degree of confidence or insignificant. This tool identifies statistically significant spatial clusters of high values (hot spots) and low values (cold spots).” So take these sentences to methods section.

Interpolation of prevalence

The sentences that stated as “Based on the sampled clusters, a Kriging interpolation was used to predict the occurrence of illnesses and care utilization”. This is also methods we applied so remove from the result and put to methods section.

Discussion last paragraph:

Rephrase the paragraph “In our interpolation exercise, the estimated occurrence of common childhood illness was high in the North, central, and Western parts of Ethiopia. Sick child health service utilisation was low in Eastern, South-western and the North-eastern parts of the county. In contrast, care utilisation was more likely high in central, Western, North-eastern parts.”

The following sentences are methods no to be considered as discussion so please remove or take it to methods section.” as Ordinary Kriging and empirical Bayesian Kriging are powerful interpolation methods that optimize the weight (28). The interpolation produced smoothed maps of sick child health service utilisation by predicting the proportion of sick child health service utilisation in the un-sampled locations (enumeration areas).”

Reviewer 4

I would like to mention the following comments:

1- Sampling: DHS must be defined.

2- Sampling: The variation between strata are different in Stratified and cluster random sampling. It needs more explanation.

3- Reliability of questionnaire needs to be mentioned.

4- Due to season differences, two-week prevalence is different for diseases. For example, diarrhoea is more in warm seasons and acute respiratory infection in cold seasons. Please clarify the exact time of year.

5- The number of children in each family might also have an effect on illnesses.

Reviewers' comments:

Reviewer's Responses to Questions

**Comments to the Author**

1. Is the manuscript technically sound, and do the data support the conclusions?

Reviewer #1: Partly

Reviewer #2: Yes

Reviewer #3: Yes

Reviewer #4: Yes

2. Has the statistical analysis been performed appropriately and rigorously? 

Reviewer #1: No

Reviewer #2: Yes

Reviewer #3: Yes

Reviewer #4: Yes

3. Have the authors made all data underlying the findings in their manuscript fully available?

Reviewer #1: No

Reviewer #2: Yes

Reviewer #3: Yes

Reviewer #4: Yes

4. Is the manuscript presented in an intelligible fashion and written in standard English?

Reviewer #1: Yes

Reviewer #2: Yes

Reviewer #3: Yes

Reviewer #4: Yes

5. Review Comments to the Author

Reviewer #1: The manuscript is very important to policy making on childhood illness and service utilization patterns which is essential for reducing child morbidity and mortality in Ethiopia. However, it is advisable for authors to revise the result section which has a lot of inconsistency in the analysis method and the presentation of results. I am concerned that great piece of work has not provided sufficiency in the results presented to demonstrate the kind of conclusion that has been provided. This paper will benefit greatly from revision of the methods and result section to ensure the findings are consistent.

I would expect authors to use the Ethiopia DHS 2020 for the revised submission of this manuscript

Reviewer #2: Overcoming the barriers to reduce infant mortality in sub Saharan Africa is of outmost importance. This work is a much needed contribution. Poverty and distance to the clinic were the major factors that affected service utilization according to this this study.

Question is ..how exactly did you measure maternal literacy what level of education did the mother have to be considered literate?

Reviewer #3: This is a very important study. Kindly pay attention to every comment and everything highlighted, underlined, or struckthrough in the attached reviewed manuscript. Hover the cursor on the highlight, underline, or strikethrough to see more comments. Kindly copy them out and address them.

Most of the statements under Results and Discussion sections are marked to be rephrased. They should be transferred to the appropriate sub-heading under Materials and Methods.

For the interpolation exercise under discussion, kindly state the implication of the results.

Reviewer #4: I would like to mention the following comments:

1- Sampling: DHS must be defined.

2- Sampling: The variation between strata are different in Stratified and cluster random sampling. It needs more explanation.

3- Reliability of questionnaire needs to be mentioned.

4- Due to season differences, two-week prevalence is different for diseases. For example, diarrhoea is more in warm seasons and acute respiratory infection in cold seasons. Please clarify the exact time of year.

5- The number of children in each family might also have an effect on illnesses.

Good Luck

6. PLOS authors have the option to publish the peer review history of their article (what does this mean?). If published, this will include your full peer review and any attached files.

Reviewer #1: No

Reviewer #2: No

Reviewer #3: No

Reviewer #4: **Yes: **Masoud Amiri

---

## [Author Response · Author response to Decision Letter 0]

12 Sep 2022

Point-by-point repones 

Dear Dr Goshu Atalay, Muluneh

 Academic Editor

 PLOS ONE

Thanks for the comments, concerns, and revision points you and the three reviewers have provided to this manuscript titled “Spatial distributions of common childhood illnesses, healthcare utilisation and associated factors in Ethiopia: Evidence from 2016 Ethiopian Demographic and Health Survey”

” number PONE-D-22-12369. 

We have now addressed the comments and amended the manuscript, you could find in the files “revised manuscript track changes” highlighted in red, the clean version-manuscript and, response to the reviewers. And we have revised the manuscript according to the PLOS ONE style and guideline and captions are corrected. 

Regarding the funding, we would like to disclose that no fund was obtained for this study and would like to be written in the manuscript in the following way. “The authors received no specific funding for this work.” 

The data availability could be described like this: - The data are available upon request from the DHS measure website but are not available in public repository because the original data set is owned by DHS measure not by the authors. Any one may contact DHS measure at archive@measuredhs.com to request the data.

The maps are created by the Authors themselves using the ArcMap and Arc toolbox menu on the ArcGIS version 10.5 (ESRI, Redlands, CA, USA). The shapefiles are freely accessed using the fowling link: https://africaopendata.org/dataset/ethiopia-shapefiles. 

Kind regards

Atkure Defar

Corresponding Author 

 

Academic editor’s comment

1. PLOS ONE requires data used for this scientific work. Even though you used freely available national data, you are requested to attach the specific data set you used for this analysis.

RESPONSE

Thank you for the concern. We have no right to share the data, However, the data are available upon request from the DHS measure website but are not available in public repository because the original data set is owned by DHS measure not by the authors. Any one may contact DHS measure at archive@measuredhs.com to request the data.

2. Add key-words from the abstract

RESPONSE

Thank you. We have now included the key words (such as; Ethiopia, Child, Spatial, Utilisation) in the abstract

3. We saw flaws, and different grammatical and typological errors, hence language edition with experts is required.

RESPONSE

Thank you for the comment. We have now made intense language edition on the manuscript.

4. Please strictly follow the PLOS ONE journal guideline (insert line number for the manuscript, add key words…) 

RESPONSE

Thank you. We have now inserted line number in the manuscript and the key words in the abstract.

5. We note you have included a table to which you do not refer in the text of your manuscript. Please ensure that you refer to Table 5 in your text; if accepted, production will need this reference to link the reader to the Table.

RESPONSE

Thank you for the comment. We have now checked this and found there is a reference site for table 5 (Page line# ).

 

Reviewer #1: 

- I would expect authors to use the Ethiopia DHS 2020 for the revised submission of this manuscript

There is a 2020 EDHS report, authors should use more recent data for analysis as changes might of occurred in the country that might have improved or increased prevalence of childhood diseases and utilization of services

RESPONSE

Thank you for the comment. This suggestion was also our interest, however, the EMiniDHS 2019 didn’t assess child health morbidity and health care utilization. The recent national level data that contains about child health care utilisation as EDHS2016. Hopefully, this result of this study will also be used to compare the upcoming EDHS 2021.

- The section on study design was used to describe Ethiopia instead of telling us the type of study, the target population. The study would benefit from a revision of this section.

RESPONSE

Thank you for the comment. We have revised the section to indicate the study design. And target population. 

- What do the authors mean by “probability proportional to segment size”.

RESPONSE

Thank you for the questions. We have now provided description about the probability proportional to segment size in the method section. It’s the process of selecting one segment when there is large sized enumeration area, which include more than 300 households. During the process of selecting parts of this EA , first that large EA should be split in to two or three segments/parts then one segment will be selected to represent the EA. It means that - in the process of selecting part of the enumeration area, proportion of the household that each segment has will be considered. (Page Line#) 

- “A total of 10,417 caregivers or mothers of under-five children were interviewed” contradicts the sample size information in the abstract and the result section “10,641”

RESPONSE

Thank you for the questions. The 10, 417 is the caregiver who were interviewed for their under-five children but the 10,641 is under-five children included in the analysis. This mean there were caregivers who had more than one child. 

- “Sick” or “Seek”. I observed the use of the word sick a couple of times. Please correct the word as appropriate

RESPONSE

Thank you so much for the comment. We have checked the use of “Seek” & “Sick” found none to be corrected. However, we used “seek” to indicated the demand/interest and “sick” is for illness. 

- The term “the two-week prevalence of common childhood illnesses “is not clear. Am not sure this is the term used in the DHS report. Authors need to ensure consistency of terms compared to the EDHS report.

RESPONSE

Thank you for the concern. The EDHS used the prevalence of each two-week illness but this manuscript considered the three illness (Fever, Diarrhoea, and suspected pneumonia) and connected with OR. That is the difference. The ways of analysis is standard and can be used to indicated common childhood illnesses. 

- “Table 1 shows the outcomes and potential predictors for health service utilization”. Table 1 didn’t differentiate between outcomes and predictors. Update table 1 to contain all these variables “age and sex of the child, woman-headed household, age of the mother or caregiver, mothers’ or caregivers’ education level, household size, wealth index, and distance to health facility”.

RESPONSE

Thank you so much the comment. We have not added a column and differentiate which is outcome OR explanatory.

- This statement “The 94 clusters without sick children and 18 clusters from Somali region without shape files were excluded in the spatial analysis. Therefore, the results of the spatial analysis are based on 531 clusters” however it should be a part of methodology as a limitation of the study

RESPONSE

Thank you so much for the comment. We have not included the issue of missing clusters shapefiles as a limitation under Strength and limitation section. 

- In the result section, I had expected to see the prevalence of each of the childhood disease which were mentioned in the methodology section however am seeing a description of the prevalence. Authors should endeavor to present the prevalence of the childhood disease

RESPONSE

- “*lack adequate number of primary sampling units or observation per stratum to estimate the standard error to get the 95% CI for acute respiratory infection and diarrhoea”. Are you saying only one primary sampling unit had people seeking health care? I think authors should endeavor to check this analysis to ensure accuracy.

RESPONSE

- Table 4 seems to be a description of measures. I believe this should be a part of the methodology section

RESPONSE

Thank you for the comment. Table 4 presented cluster and outlier analysis using Anselin Local Morans I for childcare utilisation and we believe that this could show readers where and how many clusters are with higher service utilization and low childcare utilization. Thus, this must be presented in the result section. 

- The figure 1 maps aren’t correct. Prevalence is percentage vales and should represent an area. Dot distribution analysis can’t be used to represent a prevalence analysis. Authors should replot the maps

RESPONSE

Thank you for the comment. Proportion/prevalence can be a point or vector data, we preferred these ways of presentation just to show the reader the prevalence in specific areas by using colour as an identifier. Had it been the clusters were vector we would have a better visualization. But we clusters are represented by point not vector. 

- Figure 2 don’t represent spatial patterns. Please change the distribution curves to maps. This analysis is not properly carried out

Thank you for the concern. We bring the Figure 2 from the spatial auto-corelation analysis output to show that whether the prevalence id common childhood illness and service utilization has been spatially clustered or not. This a decision for Dispersion, Clustered and random. We believe that is a good one to include in the manuscript. 

Reviewer #2: 

Overcoming the barriers to reduce infant mortality in sub Saharan Africa is of outmost importance. This work is a much needed contribution. Poverty and distance to the clinic were the major factors that affected service utilization according to this this study.

Question is ..how exactly did you measure maternal literacy what level of education did the mother have to be considered literate?

RESPONSE

Thank you for the comment. Knowledge was measure using questions that asks the education level of the mothers and the scale was analysed using the education system that Ethiopia have. 

Reviewer #3: 

-Most of the statements under Results and Discussion sections are marked to be rephrased. They should be transferred to the appropriate sub-heading under Materials and Methods.

Thank you for the comment. We have now revised the manuscript according to your suggestion. 

-For the interpolation exercise under discussion, kindly state the implication of the results.

Thank you for the comment. We have included the implcion of the interpolation result. 

-Title: From the title remove S from the word Distributions, it should be distribution

RESPONSE

Thank you for the comment. We have now removed “S” form the word Distributions in the title. 

-Abstract:

-Methods: From the second line “A total of 10,641 children under the age of five years were included in this analysis” so remove “the age of”

RESPONSE

Thank you for the comment. We have now changed the wording order. 

-Change the phrase interpolation kriging by Kriging interpolation

RESPONSE

Thank you for the comment. We have now changed the wording order “Kriging interpolation”

-Background

At the fourth sentence: As we previously have shown, mothers of sick children experience cultural, social, and poverty-related barriers to care-seeking. Replace this sentence by “As shown in our previous paper, mothers of sick children experience cultural, social, and poverty-related barriers to care-seeking”

RESPONSE

Thank you for the comment. We have now revised the writing style. 

Methods

-Sampling: From the 1st sentence please remove the comma and phrase “which we”

RESPONSE 

Thanks you and we have corrected this in the manuscript (Page. Line# )

-Outcome measurement section: replace the phrase “In the survey, the health status was assessed by the answering to the question to the mother:” by In the survey, the health status was assessed by answering the question to the mother:

RESPONSE

Thank you for the comment. We have now revised the grammatic errors and related issues in the section.

-Data management part third paragraph 1st sentence replace the phrase “Spatial heterogeneity of significant high or low of common” by Spatial heterogeneity of significantly high or low common..)

RESPONSE

Thank you for the comment. We have now revised the grammatic errors in this section (page Line#).

Result

-The 1st sub heading please replace the word characteristic by characteristics

RESPONSE

Thank you for the comment. We have now replaced the word you suggested (Page Line #). 

-As you know the result section should focus on what researchers found while the methods elaborate on what we did or what procedures we used. The following sentences should be part of methods not results section “Using hot spot analysis (Getis-Ord Gi*), we classified each cluster as being part of a spatial clustering with hotspot, cold spot at 90%, 95%, or 99% degree of confidence or insignificant. This tool identifies statistically significant spatial clusters of high values (hot spots) and low values (cold spots).” So take these sentences to methods section.

RESPONSE

Thank you for the comment. We have now deleted this description from the result section. We have enough description about it in the method section and didn’t take this to the method section,

-Interpolation of prevalence

-The sentences that stated as “Based on the sampled clusters, a Kriging interpolation was used to predict the occurrence of illnesses and care utilization”. This is also methods we applied so remove from the result and put to methods section.

RESPONSE

Thank you for the comment. We have removed this detailed methodological description from the result section. 

Discussion last paragraph:

-Rephrase the paragraph “In our interpolation exercise, the estimated occurrence of common childhood illness was high in the North, central, and Western parts of Ethiopia. Sick child health service utilisation was low in Eastern, South-western and the North-eastern parts of the county. In contrast, care utilisation was more likely high in central, Western, North-eastern parts.”

RESPONSE

Thank you for the comment. We have now revised according to your suggested. 

-The following sentences are methods no to be considered as discussion so please remove or take it to methods section.” as Ordinary Kriging and empirical Bayesian Kriging are powerful interpolation methods that optimize the weight (28). The interpolation produced smoothed maps of sick child health service utilisation by predicting the proportion of sick child health service utilisation in the un-sampled locations (enumeration areas).”

RESPONSE

Thank you for the comment. We have now revised according to your suggested (Page-- Line#--) 

Reviewer 4

1- Sampling: DHS must be defined.

RESPONSE

Thank you for the comment. We have now defined DHS (Page Line#).

2- Sampling: The variation between strata are different in Stratified and cluster random sampling. It needs more explanation.

RESPONSE

Thank you for the comment. We have now provided more explanation about the strata with respect to our sampling strategy. Strata is at higher level we used to indicate regions, but the sampling used to select the enumeration areas was a cluster random sampling. This discretion is clearly indicated in the sampling section (page line#)

3- Reliability of questionnaire needs to be mentioned.

RESPONSE

Thank you for the concern. We have now mentioned few things about the reliability of the questionnaire (page line#).

4- Due to season differences, two-week prevalence is different for diseases. For example, diarrhoea is more in warm seasons and acute respiratory infection in cold seasons. Please clarify the exact time of year.

RESPONSE

Thank you for the concern. We have now clarified the timing of the data collection and try to relate with the disease seasonality in the discussion section, alos we indicated the data collection period under method section (page line# ). As we have described we applied a secondary data analysis, we didn’t collect the data ourselves. 

5- The number of children in each family might also have an effect on illnesses.

RESPONSE

Thank you for the concern. We have included few descriptions about the effect of the number of children in the households on the outcome of the study. 

---

## [Decision Letter · Decision Letter 1]

18 Nov 2022

PONE-D-22-12369R1Spatial distribution of common childhood illnesses, healthcare utilisation and associated factors in Ethiopia: Evidence from 2016 Ethiopian Demographic and Health SurveyPLOS ONE

Dear Dr. Defar,

Thank you for submitting your manuscript to PLOS ONE. After careful consideration, we feel that it has merit but does not fully meet PLOS ONE’s publication criteria as it currently stands. Therefore, we invite you to submit a revised version of the manuscript that addresses the points raised during the review process.

We look forward to receiving your revised manuscript.

Kind regards,

Rajesh Raushan, PhD

Academic Editor

PLOS ONE

Journal Requirements:

Reviewers' comments:

Reviewer's Responses to Questions

**Comments to the Author**

1. If the authors have adequately addressed your comments raised in a previous round of review and you feel that this manuscript is now acceptable for publication, you may indicate that here to bypass the “Comments to the Author” section, enter your conflict of interest statement in the “Confidential to Editor” section, and submit your "Accept" recommendation.

Reviewer #2: All comments have been addressed

Reviewer #3: (No Response)

2. Is the manuscript technically sound, and do the data support the conclusions?

Reviewer #2: Yes

Reviewer #3: Yes

3. Has the statistical analysis been performed appropriately and rigorously? 

Reviewer #2: Yes

Reviewer #3: N/A

4. Have the authors made all data underlying the findings in their manuscript fully available?

Reviewer #2: Yes

Reviewer #3: Yes

5. Is the manuscript presented in an intelligible fashion and written in standard English?

Reviewer #2: Yes

Reviewer #3: Yes

6. Review Comments to the Author

Reviewer #2: The authors spent time addressing the questions asked apart from the data which is not available on public domain but is otherwise accessible

Reviewer #3: Thank you for the great improvement from the previous version. Kindly address line 220 [Therefore, the results of the spatial analysis are based on 531 clusters.] It should be ... were based on ...

One other suggestion was not addressed - repetition. Lines 288-297 has repeated phrases/statement to lines 310-314. Harmonize these paragraphs. Make them one paragraph.

7. PLOS authors have the option to publish the peer review history of their article (what does this mean?). If published, this will include your full peer review and any attached files.

Reviewer #2: No

Reviewer #3: No

---

## [Author Response · Author response to Decision Letter 1]

1 Jan 2023

- There is a 2020 EDHS report, authors should use more recent data for analysis as changes might of occurred in the country that might have improved or increased prevalence of childhood diseases and utilization of services

Thank you for the suggestion. The latest Ethiopian DHS data is the Ethiopian minDHS 2019, but this doesn’t include two-week childhood illness. We have used the recent 2016 Ethiopian Demographic Health Survey that contains the childhood two -weeks morbidity. 

- The section on study design was used to describe Ethiopia instead of telling us the type of study, the target population. The study would benefit from a revision of this section.

Thank you for the comment. We have checked and found that this section contains, the study design setting and period of the study. We have a bit detailed description of Ethiopia. This is because of the nature of the subject we are dealing. Spatial analysis needs to have a description of the areas or settings.

- What do the authors mean by “probability proportional to segment size”.

Thank you so much for the clarity questions. The probability proposition to segment is the method of taking a segment of the given clustered when the households in the clustered in larger than the standard. The normal one is up to 150 households. But when the clustered contains 350 households, segmentation should be done. In this case, the segmentation should be into three then taking one of the segments by considering the probability proportional to the three segments.

- “A total of 10,417 caregivers or mothers of under-five children were interviewed” contradicts the sample size information in the abstract and the result section “10,641”

Thank you for the observation. We have checked and now the sample size is consistent in both sections of the manuscript. 

- “Sick” or “Seek”. I observed the use of the word sick a couple of times. Please correct the word as appropriate

Thank you for the comment. We have now checked for all and they are corrected. 

- The term “the two-week prevalence of common childhood illnesses “is not clear. Am not sure this is the term used in the DHS report. Authors need to ensure consistency of terms compared to the EDHS report.

Thank you for the comment. We have now checked the consistency of the terms and we found that DHS is using the two-week reported symptom for each illness. We have used the very common three illnesses that need special attention during childhood development and the WHO call them common childhood illnesses. Symptoms of ARI, fever as a symptom of malaria, and diarrhea together are named common childhood illnesses. 

- “Table 1 shows the outcomes and potential predictors for health service utilization”. Table 1 didn’t differentiate between outcomes and predictors. Update table 1 to contain all these variables “age and sex of the child, woman-headed household, age of the mother or caregiver, mothers’ or caregivers’ education level, household size, wealth index, and distance to health facility”. 

Thank you for the comment. The table is updated and included the suggested variables with their details. 

- This statement “The 94 clusters without sick children and 18 clusters from Somali region without shape files were excluded in the spatial analysis. Therefore, the results of the spatial analysis are based on 531 clusters” however it should be a part of methodology as a limitation of the study

Thank you for the comment. We have now checked, and the strength and limitation section included this point. 

- In the result section, I had expected to see the prevalence of each of the childhood disease which were mentioned in the methodology section however am seeing a description of the prevalence. Authors should endeavor to present the prevalence of the childhood disease

Thank you for the comment. We have now checked that the prevalence of each childhood illness and care utilization for the symptoms, these have been presented in table 3 and described in the result section. 

- “*lack adequate number of primary sampling units or observation per stratum to estimate the standard error to get the 95% CI for acute respiratory infection and diarrhoea”. Are you saying only one primary sampling unit had people seeking health care? I think authors should endeavor to check this analysis to ensure accuracy.

Thank you for the comment. We are saying that to calculate the 95% confidence interval there are a minimum level of the proportion of the outcome in each cluster that is adequate to estimate the 95% confidence interval. There are a few strata that didn’t fulfill this which resulted in difficulty to estimate the 95% CI.

- Table 4 seems to be a description of measures. I believe this should be a part of the methodology section.

Thank you for the comment. Table 4 is the result of the analysis to identify how many clusters are high, low, or non-significant levels of common childhood illnesses and care utilization based on the Getis-Ord Gi*-statistical analysis. Thus, as a spatial analyst, we believe that this should be part of the result section. 

- The figure 1 maps aren’t correct. Prevalence is percentage vales and should represent an area. Dot distribution analysis can’t be used to represent a prevalence analysis. Authors should replot the maps

Thank you for the comment. Figure 1 intention is to show the prevalence of common childhood illnesses in each cluster. Yes, prevalence is a percentage value that can be displayed over a map using different symbology techniques in spatial visualization. We have used one of the techniques that help to show the differences using colors. 

- Figure 2 don’t represent spatial patterns. Please change the distribution curves to maps. This analysis is not properly carried out

Thank you for the comment. Figure 2 Is just to show whether there is a clustering of common childhood illnesses in the specified geographic areas. That is the result of the spatial autocorrelation analysis. Figure 1 shows a simple description but figure 2 shows the distribution is dispersed, random, or clustered. We believe that this is important to include in the manuscript.

---

## [Decision Letter · Decision Letter 2]

27 Jan 2023

Spatial distribution of common childhood illnesses, healthcare utilisation and associated factors in Ethiopia: Evidence from 2016 Ethiopian Demographic and Health Survey

PONE-D-22-12369R2

Dear Dr. DEFAR,

We’re pleased to inform you that your manuscript has been judged scientifically suitable for publication and will be formally accepted for publication once it meets all outstanding technical requirements.

Additional Editor Comments (optional):

Thank you for addressing the relevant comments and revised manuscript accordingly in well scientific manner. When looking for explanatory variables regulating treatment seeking behaviour you have considered in the OLS model, household economic status and distance found significant are consistent with the previous study. But, the mother's education found insignificant possibly due to small sample size of literate mothers in some of the clusters, as most of the studies have found mother education as a strong regulating factors for health services utilization.

Just check for grammatical editing and follow requirement of PLOS ONE publication process.

Kind regards,

Rajesh Raushan, PhD

Academic Editor

PLOS ONE

---

## [Editor Report · Acceptance letter]

24 Feb 2023

PONE-D-22-12369R2 

Spatial distribution of common childhood illnesses, healthcare utilisation and associated factors in Ethiopia: Evidence from 2016 Ethiopian Demographic and Health Survey 

Dear Dr. Defar:

I'm pleased to inform you that your manuscript has been deemed suitable for publication in PLOS ONE. Congratulations! Your manuscript is now with our production department. 

Kind regards, 

on behalf of

Dr. Rajesh Raushan 

Academic Editor

PLOS ONE